# Identification of a transporter complex responsible for the cytosolic entry of nitrogen-containing bisphosphonates

Zhou Yu[1,2], Lauren E Surface[1,3,4,5], Chong Yon Park[6], Max A Horlbeck[5,6,7], Gregory A Wyant[5,8,9,10,11], Monther Abu-Remaileh[5,8,9,10,11], Timothy R Peterson[12], David M Sabatini[5,8,9,10,11], Jonathan S Weissman[5,6,7], Erin K O'Shea[1,2,3,4,5]*

[1]Department of Molecular and Cellular Biology, Harvard University, Cambridge, United States; [2]Janelia Research Campus, Howard Hughes Medical Institute, Ashburn, United States; [3]Faculty of Arts and Sciences Center for Systems Biology, Harvard University, Cambridge, United States; [4]Department of Chemistry and Chemical Biology, Harvard University, Cambridge, United States; [5]Howard Hughes Medical Institute, Bethesda, United States; [6]Department of Cellular and Molecular Pharmacology, University of California, San Francisco, San Francisco, United States; [7]Center for RNA Systems Biology, University of California, San Francisco, San Francisco, United States; [8]Whitehead Institute for Biomedical Research, Cambridge, United States; [9]Department of Biology, Massachusetts Institute of Technology, Cambridge, United States; [10]Koch Institute for Integrative Cancer Research, Cambridge, United States; [11]Broad Institute of MIT and Harvard, Cambridge, United States; [12]Division of Bone & Mineral Diseases, Department of Genetics, Institute for Public Health, Washington University School of Medicine, St. Louis, United States

*For correspondence: osheae@hhmi.org

**Abstract** Nitrogen-containing-bisphosphonates (N-BPs) are a class of drugs widely prescribed to treat osteoporosis and other bone-related diseases. Although previous studies have established that N-BPs function by inhibiting the mevalonate pathway in osteoclasts, the mechanism by which N-BPs enter the cytosol from the extracellular space to reach their molecular target is not understood. Here, we implemented a CRISPRi-mediated genome-wide screen and identified *SLC37A3* (solute carrier family 37 member A3) as a gene required for the action of N-BPs in mammalian cells. We observed that SLC37A3 forms a complex with ATRAID (all-trans retinoic acid-induced differentiation factor), a previously identified genetic target of N-BPs. SLC37A3 and ATRAID localize to lysosomes and are required for releasing N-BP molecules that have trafficked to lysosomes through fluid-phase endocytosis into the cytosol. Our results elucidate the route by which N-BPs are delivered to their molecular target, addressing a key aspect of the mechanism of action of N-BPs that may have significant clinical relevance.
DOI: https://doi.org/10.7554/eLife.36620.001

## Introduction

N-BPs are the most commonly prescribed drugs used to treat osteoporosis (*Drake et al., 2008*). They have two negatively charged phosphonate groups that bind to hydroxyapatite crystals with high affinity and enable efficient accumulation of N-BPs on the bone surface (*Drake et al., 2008*). Osteoclasts, the major cell type responsible for bone resorption, release N-BPs from the bone matrix by dissolving bone mineral and then take up N-BPs through fluid-phase endocytosis (*Drake et al.,*

**eLife digest** As some people age, their bones may become weak, brittle, and break easily. This condition is called osteoporosis. To treat osteoporosis, doctors often prescribe drugs called nitrogen-containing bisphosphonates (NBPs). These drugs destroy cells called osteoclasts, which break down bone. This helps restore bone mass. To kill osteoclasts, the drugs must enter these cells. First, they must pass through an oily protective layer called a membrane. It is not completely clear how NBPs, which prefer to stay in water-like environments, can cross this oily membrane and enter osteoclasts.

Understanding how NBPs cross the membrane is important to ensure the drugs work effectively. If NBPs do not efficiently cross the membrane, they will not work properly and may cause harmful side effects. Many patients who take NBPs suffer from side effects such as abnormal fractures.

Now, Yu et al. show that two proteins help NBPs cross the membrane. In the experiments, proteins were removed from human cancer cells one at a time using a technique called CRISPRi. CRISPRi enabled the researchers to systematically turn off the genes for each protein and track what affect this had on the NBPs' ability to cross the membrane. When one of the two genes called SLC37A3 and ATRAID was turned off, NBPs could not get into cells. The protein produced by the SLC37A3 gene opens a gate in the cell membrane allowing NBPs to enter osteoclasts. The protein made by the ATRAID gene helps this gate protein, and without it, the SLC37A3 proteins are unstable and NBPs cannot enter.

Some people have variations of the SLC37A3 and ATRAID genes. Testing whether these genetic variations may alter NBPs' ability to cross the membrane of osteoclasts in mice, might one day help physicians predict which patients with have side effects.

DOI: https://doi.org/10.7554/eLife.36620.002

*2008*; *Thompson et al., 2006*). N-BPs subsequently inhibit farnesyl diphosphate synthase (FDPS) in the mevalonate pathway and reduce protein prenylation, an essential post-translational lipid modification required for the function of numerous proteins such as Ras, Rab and Rho, thereby inducing apoptosis in osteoclasts and diminishing their bone-resorption activities (*Drake et al., 2008*; *Dunford et al., 2001*; *Fisher et al., 2000*; *Hughes et al., 1995*; *Kavanagh et al., 2006*; *Luckman et al., 1998a*; *Luckman et al., 1998b*; *van Beek et al., 1999*). However, it is not known how highly charged N-BPs exit the endocytic pathway to target FDPS, which is localized to the cytosol and peroxisomes (*Martín et al., 2007*). It has been proposed that the acidification of endocytic compartments might neutralize the negative charges on the phosphonate groups and allow N-BPs to diffuse across the vesicle membrane (*Thompson et al., 2006*), but this model does not address the issue that the amine groups in N-BPs become positively charged in acidic environments. An alternative model is that a transporter exists that facilitates the exit of N-BPs from endocytic vesicles.

## Results and discussion

To gain further insight into the mechanism of action of N-BPs, including the mechanism by which N-BPs are delivered to their molecular target, we implemented an unbiased genome-wide screening approach based on CRISPR-mediated interference (CRISPRi) (*Figure 1A*) (*Gilbert et al., 2014*). We transduced a genome-scale CRISPRi single-guide RNA (sgRNA) library into K562 human myeloid leukemia cells that stably express a dCas9-KRAB fusion protein, which functions as an sgRNA-guided transcription inhibitor. The cells were split into a population treated with alendronate (ALN), a representative N-BP, and an untreated control population. Through deep sequencing, we quantified the enrichment/depletion of each sgRNA in the treated population compared to the control population (*Figure 1B*, *Figure 1—figure supplement 1A*), and designated the target genes of those sgRNAs enriched in the treated population as resistance hits and those depleted as sensitizing hits (*Figure 1B*, *Figure 1—figure supplement 1B* and *Supplementary file 1*). Consistent with the current model for the action of N-BPs, enzymes, co-factors, and regulators of the mevalonate pathway are enriched in top hits (*Figure 1C–D* and *Figure 1—figure supplement 1C–D*). Particularly, in accordance with the model that N-BPs induce cell death through inhibiting the enzymatic activities

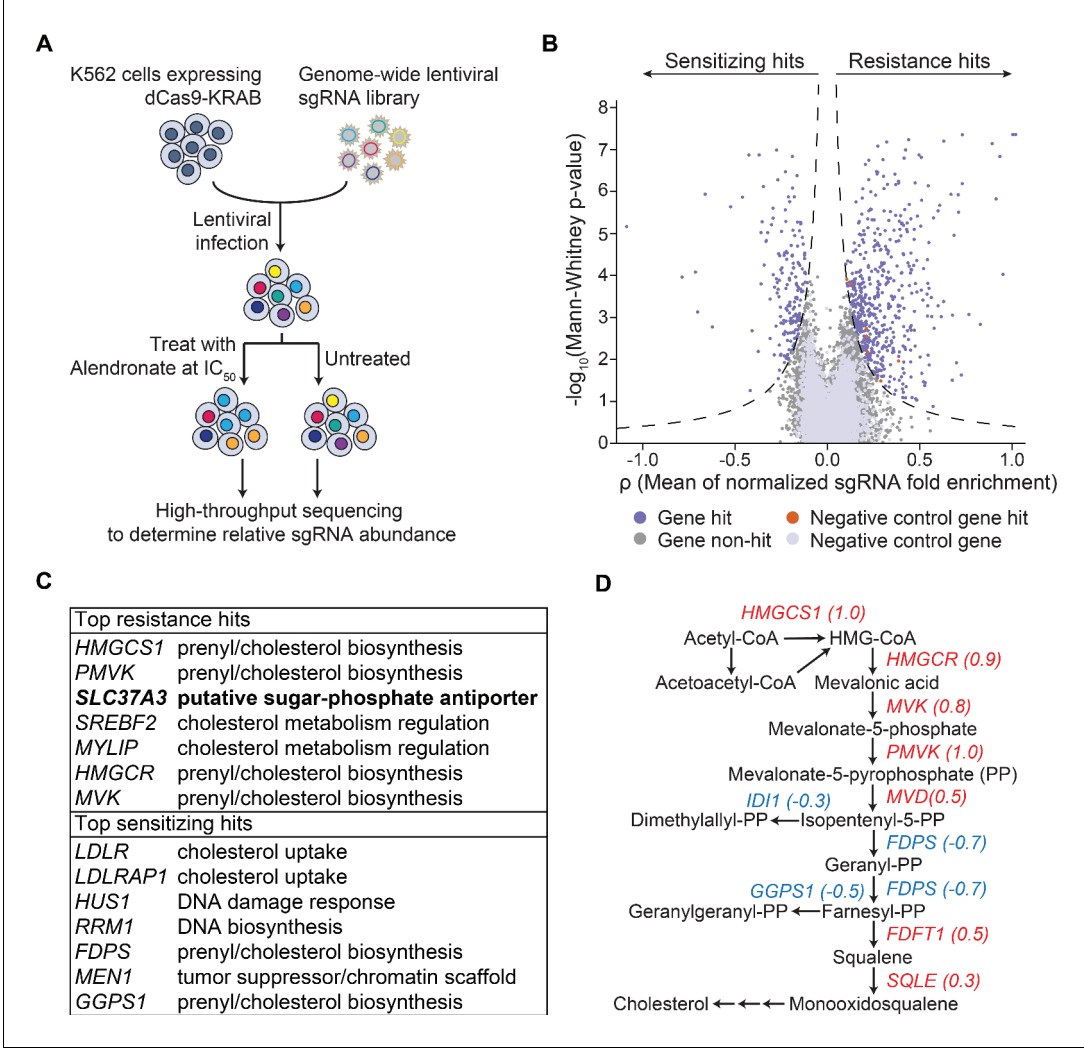

**Figure 1.** An unbiased CRISPRi screen identifies genetic targets of alendronate. (**A**) Schematic illustrating the workflow of the genome-wide CRISPRi screen. The IC$_{50}$ of alendronate in K562 cells is 250 μM. (**B**) Volcano plot showing, for each gene, a ρ score that averages the normalized fold enrichment (in the treated population compared to the untreated control) of the gene's three most effective sgRNAs, and a Mann-Whitney *P*-value for fold enrichment (*Gilbert et al., 2014*). The dashed lines represent thresholds used to identify significant hits. Positive ρ scores correspond to resistance hits and negative scores to sensitizing hits. (**C**) Gene names and annotated functions of the top seven resistance and sensitizing hits. Genes are sorted by the absolute values of their ρ scores in descending order. *SLC37A3* is marked in bold. (**D**) Diagram of the mevalonate pathway, with genes in the pathway that were identified as significant hits marked with their ρ scores. Resistance hits are color-coded in red and sensitizing hits in blue.

DOI: https://doi.org/10.7554/eLife.36620.003

The following figure supplement is available for figure 1:

**Figure supplement 1.** Additional analysis of the whole-genome CRISPRi screen.

DOI: https://doi.org/10.7554/eLife.36620.004

---

of FDPS and GGPPS1 (*Drake et al., 2008*), silencing of *FDPS* and *GGPPS1* strongly sensitized cells to ALN (*Figure 1C–D*). However, in contradiction with the current model of N-BP action, we observed that silencing of numerous enzymes in the pathway upstream of *FDPS* in fact conferred strong resistance to ALN (*Figure 1D*). A recent genome-wide genetic interaction study may resolve this paradox (Horlbeck et al., unpublished). That work demonstrated that isopentenyl-5-pyrophosphate (IPP), the substrate of FDPS, is a toxic intermediate that interferes with DNA synthesis and

causes DNA damage, suggesting that inhibition of enzymes upstream of *FDPS* protects cells from ALN by preventing ALN-induced accumulation of IPP.

Amongst the resistance hits not known to be involved in the mevalonate pathway, the gene that conferred the strongest resistance was *SLC37A3* (*Figure 1C*), which is predicted to encode a membrane protein with 12 transmembrane segments (*Chou et al., 2013*). *SLC37A3* also appeared as a top resistance hit in a second CRISPRi screen using zoledronate, another representative N-BP, as the selection agent (*Figure 1—figure supplement 1E* and *Supplementary file 2*), further supporting its role in the mechanism of action of N-BPs. SLC37A3 is predicted based on sequence homology to be a glucose-6-phosphate/phosphate antiporter (*Chou et al., 2013*). However, it has been demonstrated that SLC37A3 in fact lacks this predicted activity (*Pan et al., 2011*). To the best of our knowledge, the physiological function of SLC37A3 has remained elusive. Intriguingly, a recent human protein interactome study reported an interaction between SLC37A3 and ATRAID (*Huttlin et al., 2017*), a type I transmembrane protein that was identified as an N-BP target in a previous work (Surface et al., unpublished) and our zoledronate CRISPRi screen (*Figure 1—figure supplement 1E*), suggesting SLC37A3 and ATRAID might be functionally related.

To investigate the roles of *SLC37A3* and *ATRAID* in the action of N-BPs, we generated *SLC37A3*-knockout (*SLC37A3*KO) and *ATRAID*-knockout (*ATRAID*KO) cells in K562 cells, human embryo kidney (HEK) 293 T cells and murine macrophage-like RAW 264.7 cells (*Figure 2—figure supplement 1A–E*) (Surface et al., unpublished; *Ran et al., 2013*), with K562 and HEK 293 T cells serving as human cell models that represent distinct lineages, and RAW 264.7 macrophages as a mouse cell model that can be differentiated into mature osteoclasts (*Collin-Osdoby and Osdoby, 2012*). Consistent with the CRISPRi screen and our previous results, knockout of *SLC37A3* and *ATRAID* in all cell types conferred resistance to ALN (*Figure 2A–C*) (Surface et al., unpublished). Similarly, mature osteoclasts differentiated from knockout RAW cells are more resistant to ALN compared to those differentiated from wild-type cells (*Figure 2D* and *Figure 2—figure supplement 2A*) (Surface et al., unpublished). We also measured the reduction in protein prenylation as a readout of N-BP toxicity and verified that ALN treatment had significantly less of an effect on protein prenylation in *SLC37A3*KO and *ATRAID*KO cells compared to wild-type cells (*Figure 2E*). Complementation with epitope-tagged SLC37A3 or either isoform of ATRAID (a short isoform, UniProt Q6UW56-1, and a long isoform, UniProt Q6UW56-3) in *SLC37A3*KO or *ATRAID*KO HEK 293 T cells, respectively, restored sensitivity to ALN (*Figure 2—figure supplement 2B–D*), confirming that the resistance to ALN observed in the knockout cells is indeed caused by the lack of SLC37A3 or ATRAID expression, and that the epitope-tagged versions of the two proteins are functional.

The knockout cells are also resistant to N-BPs other than ALN (*Figure 2—figure supplement 2E*) but not to non-nitrogen-containing bisphosphonates (non-N-BPs), which do not target FDPS (*Figure 2—figure supplement 2F*) (*Drake et al., 2008*). Interestingly, knockout of *SLC37A3* and *ATRAID* did not protect cells from lovastatin (LOV), a statin drug that also targets the mevalonate pathway (*Figure 2—figure supplement 2G–I*) (*Tiwari and Khokhar, 2014*). The distinctive responses of *SLC37A3*KO and *ATRAID*KO cells to N-BPs, non-N-BPs, and LOV indicate that the roles of *SLC37A3* and *ATRAID* in the mechanism of action of N-BPs are not related to the mevalonate pathway but are instead specific to N-BPs. As knockout of *SLC37A3* and *ATRAID* in different cell types conferred similar responses, we focused on HEK 293 T cells for further studies as they host various tools for molecular biology.

To probe the epistatic relationship between the two genes, we generated double knockout (*ATRAID*KO; *SLC37A3*KO, abbreviated as KO[2]) cells (*Figure 2—figure supplement 1F*). We observed that *SLC37A3*KO cells are more resistant to ALN compared to *ATRAID*KO cells, and the knockout of *ATRAID* in addition to *SLC37A3* did not further protect cells from the drug (*Figure 2F* and *Figure 2—figure supplement 2J*), indicating that *ATRAID* and *SLC37A3* are indeed functionally related and that *SLC37A3* is epistatic to *ATRAID*.

Next, we investigated the mechanism underlying the observed functional relationship between *ATRAID* and *SLC37A3*. As protein localization can often provide clues to protein function and functionally linked proteins frequently share subcellular distribution patterns, we expressed functional, epitope-tagged SLC37A3 and ATRAID (*Figure 2—figure supplement 2B–C*) and characterized their localization with immunofluorescence (IF). We confirmed that epitope-tagged SLC37A3 and ATRAID are not over-expressed (*Figure 3—figure supplement 1A*). We observed that SLC37A3 co-localizes with LAMP2, a lysosomal marker, but not with Na$^+$/K$^+$-ATPase or EEA1, which mark the plasma

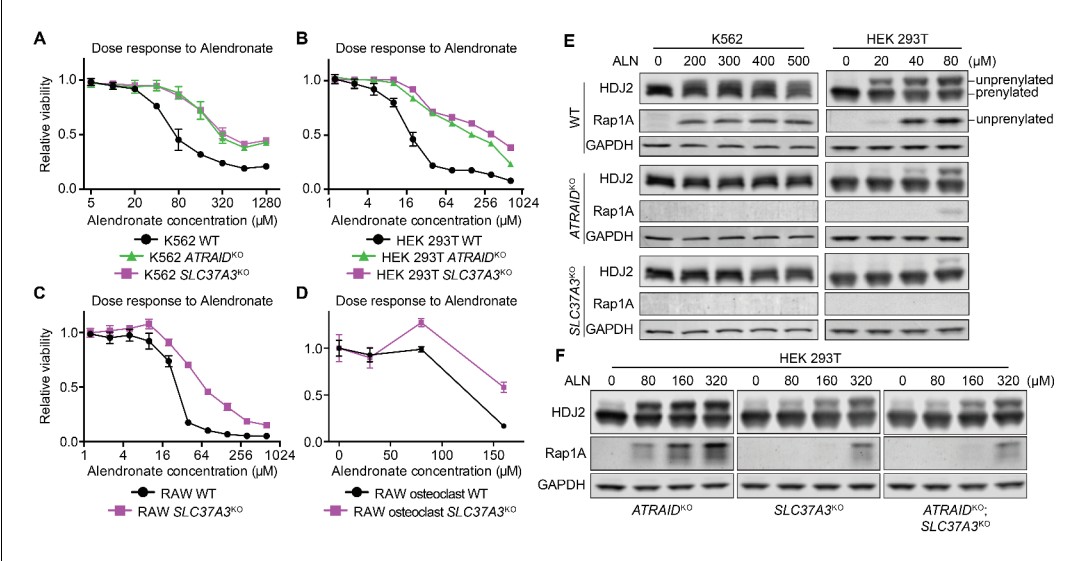

**Figure 2.** *SLC37A3* and *ATRAID* are functionally related genes required for the mechanism of action of N-BPs. (A–D) Dose response curves of wild-type, *ATRAID*KO and *SLC37A3*KO K562 cells (A) and HEK 293 T cells (B), and wild-type and *SLC37A3*KO RAW cells (both undifferentiated macrophages, (C), and differentiated osteoclasts, (D) to alendronate. Cells were treated with a series of doses of alendronate (x-axis) for 48 hr. Relative cell viability was determined by measuring post-treatment total cellular ATP levels and normalizing to those in untreated cells (y-axis). Data depict mean with s.d. for biological triplicate measurements. (E) Immunoblots measuring alendronate-induced reduction in protein prenylation in wild-type and knockout K562 and HEK 293 T cells. Cells were treated with indicated doses of alendronate for 24 hr before analysis by immunoblotting. (F) Immunoblots comparing alendronate-induced reduction in protein prenylation in single and double-knockout HEK 293 T cells. Experimental procedure is as in (C). Note that higher alendronate doses were used in (F) compared to (E) to induce detectable levels of unprenylated proteins. ALN: alendronate.

DOI: https://doi.org/10.7554/eLife.36620.005

The following figure supplements are available for figure 2:

**Figure supplement 1.** Genotypes of knockout cells used in this study.
DOI: https://doi.org/10.7554/eLife.36620.006

**Figure supplement 2.** Additional evidence that validates *SLC37A3* and *ATRAID* as functionally related genes required for the mechanism of action of N-BPs.
DOI: https://doi.org/10.7554/eLife.36620.007

membrane and early endosomes, respectively (*Figure 3A,B* and *Figure 3—figure supplement 1C*). Consistent with a previous report (*Ding et al., 2015*), both isoforms of ATRAID also predominantly localize to lysosomes but not to the plasma membrane or early endosomes (*Figure 3C–D* and *Figure 3—figure supplement 1D,F–H*). When we co-expressed functional and epitope-tagged SLC37A3 and ATRAID (*Figure 3—figure supplement 1A–B*), we observed that both isoforms of ATRAID predominantly co-localize with SLC37A3 (*Figure 3E* and *Figure 3—figure supplement 1E*). To investigate whether the observed co-localization between SLC37A3 and ATRAID represents a physical interaction, we performed reciprocal co-immunoprecipitation (co-IP) experiments in HEK 293 T cells over-expressing the two proteins. We detected an interaction between SLC37A3 and ATRAID in both pull-down directions (*Figure 3F*), confirming that SLC37A3 and ATRAID physically interact, likely forming a lysosomal complex.

As it has been reported that the expression levels of certain solute carriers depend on the presence of accessory proteins (*Makrides et al., 2014*), we investigated the possibility that the functional relationship between ATRAID and SLC37A3 is due to the impaired expression of SLC37A3 in the absence of ATRAID. Indeed, when we expressed SLC37A3 in the KO[2] cells, we observed a substantial decrease in the protein level of SLC37A3 compared to that when expressed in the *SLC37A3*KO background (*Figure 3G* and *Figure 3—figure supplement 2A*), even though the mRNA levels of *SLC37A3* in both backgrounds are similar (*Figure 3—figure supplement 1A*). The protein level of SLC37A3 was restored by complementation with either isoform of ATRAID (*Figure 3G*). (Note that the reduction in SLC37A3 expression in the absence of ATRAID is not clearly observed in *Figure 3F* due to the over-expression of SLC37A3. Indeed, in samples analyzed in *Figure 3F*, SLC37A3 exists

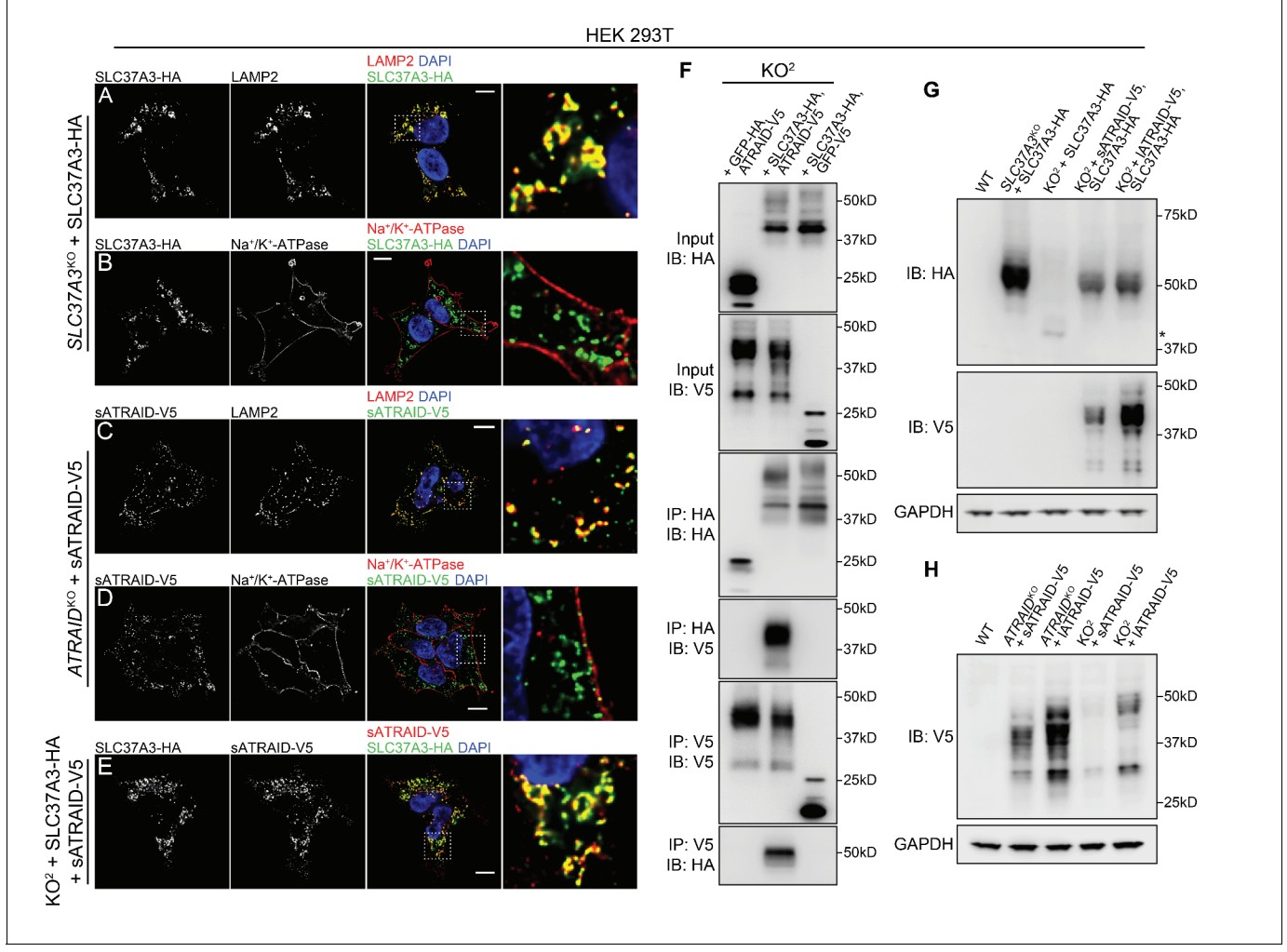

**Figure 3.** SLC37A3 and ATRAID form a lysosomal complex and are inter-dependent for their stable expression. (A–D) Localization of HA-tagged SLC37A3 (SLC37A3-HA) (A–B) and V5-tagged short isoform of ATRAID (sATRAID-V5) (C–D) shown with markers for lysosomes (LAMP2, (A and C) and the plasma membrane (Na$^+$/K$^+$-ATPase, (B and D). (E) Co-localization of SLC37A3-HA and sATRAID-V5. Nuclei were stained with DAPI in blue. Scale bars represent 10 μm. Each image displayed is the representative example chosen from at least five similar images. (F) Reciprocal co-IP of SLC37A3A3-HA and sATRAID-V5 in KO$^2$ HEK 293 T cells stably overexpressing both proteins. In each negative control cell line, one of the two tagged proteins was replaced with GFP tagged with the same epitope. (G) Immunoblots measuring SLC37A3-HA protein levels in various cells, showing that deletion of *ATRAID* significantly reduces the protein level of SLC37A3-HA. The un-glycosylated population of SLC37A3 that appears in the absence of ATRAID is marked with an asterisk. (H) Immunoblots measuring ATRAID-V5 protein levels in various cells, demonstrating that deletion of *SLC37A3* significantly reduces the protein level of ATRAID-V5. IP, immunoprecipitation. IB, immunoblot. KO$^2$: *ATRAID*$^{KO}$; *SLC37A3*$^{KO}$.

DOI: https://doi.org/10.7554/eLife.36620.008

The following figure supplements are available for figure 3:

**Figure supplement 1.** Additional evidence supporting that SLC37A3 and ATRAID form a lysosomal complex.

DOI: https://doi.org/10.7554/eLife.36620.009

**Figure supplement 2.** Additional evidence supporting that SLC37A3 and ATRAID depend on each other for stable expression.

DOI: https://doi.org/10.7554/eLife.36620.010

predominantly in an un-glycosylated form, suggesting saturation of machineries required for the post-translational processing of SLC37A3 (compare *Figure 3F and G*, also see discussion below).) In reciprocity, we also observed reduced protein levels of both isoforms of ATRAID in the KO$^2$ background that cannot be explained by changes in transcript levels (*Figure 3H*, *Figure 3—figure supplement 2B–C* and *Figure 3—figure supplement 1A*). As the translation efficiency of *SLC37A3* transcripts is not significantly altered in the absence of ATRAID (*Figure 3—figure supplement 2D–E*), the decreased SLC37A3 protein level in the KO$^2$ background is likely caused by shortened

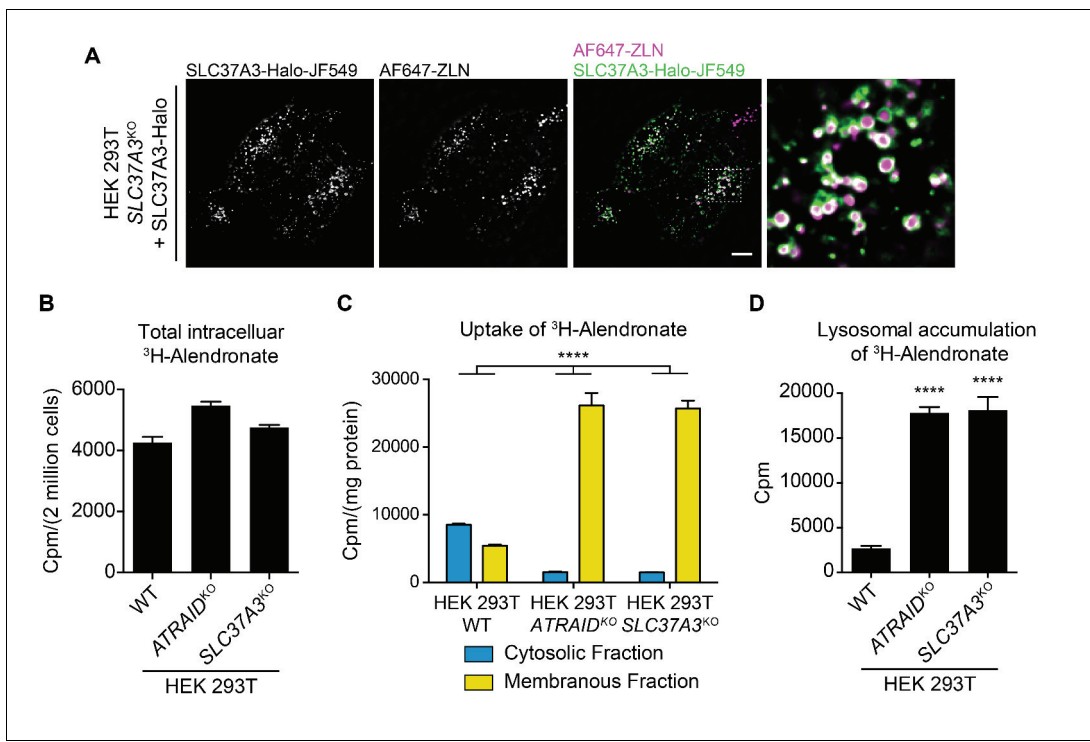

**Figure 4.** SLC37A3 and ATRAID transport N-BPs from the lumen of lysosomes into the cytosol. (**A**) Live imaging of HEK 293 T cells that express Halo tagged SLC37A3 (SLC37A3-Halo) and have internalized AlexaFlour 647 labeled zoledronate (AF647-ZLN). SLC37A3-Halo is labeled with Janelia flour 549 (JF549). SLC37A3-Halo is expressed at a lower-than-endogenous level and has been verified to be functional (data not shown). The scale bar represents 10 μm. The image displayed is a representative example chosen from five similar images. (**B**) Radioactive uptake assay measuring total intracellular radioactivity in indicated HEK 293 T cells treated with $^{3}$H-alendronate. Data depict mean and s.d. for biological triplicate measurements. (**C**) Radioactive uptake assay measuring levels of radioactivity in subcellular fractions in indicated HEK 293 T cells treated with $^{3}$H-alendronate. Data depict mean with s.d. for biological duplicate measurements. Significance was determined using unpaired two-way ANOVA test. Effect of genotype: $F_{(2,6)} = 74.93$, $p<0.0001$; effect of subcellular location: $F_{(1,6)} = 864.9$, $p<0.0001$; effect of interaction between genotype and subcellular location: $F_{(2,6)} = 312.4$, $p<0.0001$. (**D**) Radioactive uptake assay measuring levels of radioactivity in lysosomes purified from indicated HEK 293 T cells treated with $^{3}$H-alendronate. Data depict mean and s.d. for biological triplicate measurements. Significance was determined using two-tailed unpaired t-test with equal s.d. Comparison between wild-type and $ATRAID^{KO}$ cells: df = 4, t = 36.24, $p<0.0001$. Comparison between wild-type and $SLC37A3^{KO}$ cells: df = 4, t = 17.96, $p<0.0001$. HEK 293 T cells were treated with 1 μCi/mL $^{3}$H-alendronate for 24 hr in (**B–C**) and 3 hr in (**D**). ****: $p<0.0001$.

DOI: https://doi.org/10.7554/eLife.36620.011

The following figure supplement is available for figure 4:

**Figure supplement 1.** Quality controls for the radioactive uptake assays.

DOI: https://doi.org/10.7554/eLife.36620.012

protein half-life, suggesting that ATRAID and SLC37A3 are mutually dependent for their stability. Intriguingly, the deletion of ATRAID also altered the glycosylation pattern of SLC37A3. In the absence of ATRAID, the mature, glycosylated population of SLC37A3 (around 50kD) became undetectable, whereas a population of un-glycosylated SLC37A3 (around 40kD) emerged (*Figure 3G* and *Figure 3—figure supplement 2F*). Moreover, in cells overexpressing SLC37A3, although a significant proportion of SLC37A3 remained un-glycosylated, only the glycosylated population of SLC37A3 interacted with ATRAID (*Figure 3—figure supplement 2G*), suggesting that the interaction with ATRAID is crucial for the expression of correctly modified SLC37A3.

Finally, we explored the mechanism by which the knockout of *SLC37A3* and *ATRAID* conferred resistance to N-BPs. Given the predicted function of SLC37A3 as a transporter, we hypothesized that ATRAID and SLC37A3 transport N-BP molecules across the lipid bilayer to inhibit FDPS. Indeed,

we observed that SLC37A3 resides on the membrane of vesicles in which internalized fluorescently-labeled zoledronate (AF647-ZLN) accumulates (*Figure 4A*), supporting its role as a transporter of N-BPs. This hypothesis is also consistent with our finding that the role of *SLC37A3* and *ATRAID* in the mechanism of action of N-BPs is specific to the chemical properties of N-BPs. We implemented radioactive uptake assays to test this hypothesis. When we incubated wild-type, *ATRAID*[KO] and *SLC37A3*[KO] cells with radioactive ALN ([3]H-ALN) and measured total intracellular radioactivity, we observed no significant difference in whole-cell accumulation of radioactivity between the knockouts and wild-type cells (*Figure 4B*). As N-BP molecules accumulate in SLC37A3-positive vesicles and SLC37A3 localizes to lysosomes, we further hypothesized that N-BP molecules traffic to lysosomes and that SLC37A3 and ATRAID, together as a lysosomal complex, might function to release N-BP molecules from the lumen of lysosomes into the cytosol. This model predicts that the total amount of intracellular [3]H-ALN will remain the same in wild-type, *SLC37A3*[KO] and *ATRAID*[KO] cells, but [3]H-ALN will not be able to exit lysosomes in knockout cells. To detect this potential shift in the subcellular distribution of [3]H-ALN in knockout cells, we used digitonin to selectively permeabilize the plasma membrane of [3]H-ALN treated cells and generated a cytosolic fraction and a membranous fraction, which contained intact membrane-bound organelles (*Figure 4—figure supplement 1A*) (*Liu and Fagotto, 2011*). We observed that the distribution of radioactive signal changed from being primarily in the cytosolic fraction in wild-type cells to being predominantly in membranous fractions in *SLC37A3*[KO] and *ATRAID*[KO] cells (*Figure 4C*), suggesting that [3]H-ALN cannot be released from membrane-bound organelles in knockout cells. As our model specifically predicts that [3]H-ALN will be trapped in lysosomes in the absence of SLC37A3 or ATRAID, we affinity-purified lysosomes from [3]H-ALN treated cells (*Figure 4—figure supplement 1B–D*) (*Abu-Remaileh et al., 2017*; *Wyant et al., 2017*) to assess the lysosomal accumulation of [3]H-ALN. As our model predicted, we observed a significant enrichment of [3]H-ALN in lysosomes purified from knockout cells compared to those from wild-type cells (*Figure 4D*). Additionally, consistent with the observation that ATRAID is required for the stable expression of SLC37A3, *ATRAID*[KO] cells phenocopied *SLC37A3*[KO] cells in these uptake assays. Taken together, our results suggest that N-BPs traffic to lysosomes after internalization through endocytosis, and SLC37A3 and ATRAID form a lysosomal transporter complex that releases N-BP molecules from the lumen of lysosomes into the cytosol.

In summary, this study elucidates the route by which N-BPs enter the cytosol and inhibit their molecular target. As a recent study has proposed that patients who harbor a genetic variant of *GGPS1* might be more prone to the side-effects of N-BP treatment (*Roca-Ayats et al., 2017*), it is possible that patients with variants of *SLC37A3* or *ATRAID*, which are genes crucial for the action of N-BPs, might also exhibit non-canonical responsiveness to the drugs. Therefore, our results may bear significant relevance to the clinical applications of N-BPs.

# Materials and methods

## Key resources table

| Reagent type (species) or resource | Designation | Source or reference | Identifiers | Additional information |
|---|---|---|---|---|
| Gene (*Homo sapiens*) | *SLC37A3* | NA | NCBI: 84255 | |
| Gene (*Mus musculus*) | *Slc37a3* | NA | NCBI: 72144 | |
| Gene (*H. sapiens*) | *ATRAID* | NA | NCBI: 51374 | |
| Cell line (*H. sapiens*) | K562 | ATCC | ATCC: CCL-243, RRID:CVCL_0004 | |
| Cell line (*H. sapiens*) | HEK 293T | ATCC | ATCC: CRL-3216, RRID:CVCL_0063 | |
| Cell line (*M. musculus*) | RAW 264.7 | ATCC | ATCC: TIB-71, RRID:CVCL_0493 | |
| Cell line (*H. sapiens*) | K562 *SLC37A3* knock-out | this paper | | *SLC37A3* was deleted by removing exon 6 |
| Cell line (*H. sapiens*) | HEK 293T *SLC37A3* knock-out | this paper | | *SLC37A3* was deleted by removing exon 6 |

*Continued on next page*

Continued

| Reagent type (species) or resource | Designation | Source or reference | Identifiers | Additional information |
|---|---|---|---|---|
| Cell line (*H. sapiens*) | K562 *ATRAID* knock-out | this paper | | *ATRAID* was deleted by trucating exon 3, 4 and 5 |
| Cell line (*H. sapiens*) | HEK 293T *ATRAID* knock-out | this paper | | *ATRAID* was deleted by trucating exon 3, 4 and 5 |
| Cell line (*H. sapiens*) | HEK 293T KO2 (double knock-out) | this paper | | *ATRAID* was deleted in the *SLC37A3* knockout background |
| Cell line (*M. musculus*) | RAW *Slc37a3* knock-out | this paper | | *Slc37a3* was deleted by introducing microdeletions in exon 2 |
| Transfected construct (*H. sapiens*) | HEK 293T SLC37A3 KO + SLC37A3-HA | this paper | | An HA-tagged SLC37A3 CDS under PGK promoter was integrated in to the *AAVS1* expression harbor in *SLC37A3* knockout background |
| Transfected construct (*H. sapiens*) | HEK 293T ATRAIDKO + s/lATRAID-V5 | this paper | | A V5-tagged short/long ATRAID CDS under PGK promoter was integrated in to the AAVS1 expression harbor in ATRAID knockout background |
| Transfected construct (*H. sapiens*) | HEK 293T KO2 + SLC37 A3-HA | this paper | | An HA-tagged SLC37A3 CDS under PGK promoter was integrated in to the *AAVS1* expression harbor in double knockout background |
| Transfected construct (*H. sapiens*) | HEK 293T KO2 + SLC37 A3-HA + s/lATRAID-V5 | this paper | | letiviral vectors containing V5-tagged short/long ATRAID CDS under PGK promoter was transduced into KO2 + SLC37A3-HA background |
| Antibody | anti-HA (rat mAb) | Sigma-Aldrich | Sigma-Aldrich: 11867423001, RRID:AB_390918 | |
| Antibody | anti-V5 (rabbit mAb) | Cell Signaling Technology | Cell Signaling Technology: 13202, RRID:AB_2687461 | |
| Antibody | anti-V5 (mouse mAb) | ThermoFisher | ThermoFisher: R960-25, RRID:AB_2556564 | |
| Antibody | anti-LAMP2 (mouse mAb) | Santa Cruz Biotechnology | Santa Cruz: sc-18822, RRID:AB_626858 | |
| Antibody | anti-ATP1A1 (mouse mAb) | Abcam | Abcam: ab7671, RRID:AB_306023 | |
| Antibody | anti-Rap1A (goat pAb) | Santa Cruz Biotechnology | Santa Cruz: sc-1482 | This item has been discontinued due to animal welfare concerns |
| Antibody | anti-HDJ2 (mouse mAb) | ThermoFisher | ThermoFisher: MS-225-P0, RRID:AB_10982482 | |
| Antibody | anti-GAPDH (rabbit mAb) | Cell Signaling Technology | Cell Signaling Technology: 2118, RRID:AB_561053 | |
| Recombinant DNA reagent | pSpCas9(BB)—2A-GFP | Addgene | Addgene: 48138 | |
| Recombinant DNA reagent | AAVS1-Puro-PGK1 —3 × FLAG-TwinStrep | Addgene | Addgene: 68375 | |

*Continued on next page*

*Continued*

| Reagent type (species) or resource | Designation | Source or reference | Identifiers | Additional information |
|---|---|---|---|---|
| Recombinant DNA reagent | pLenti PGK Hygro DEST (w530-1) | Addgene | Addgene: 19066 | |
| Recombinant DNA reagent | AAVS1-Puro-PGK1-SLC37A3-HA | this paper | | The FLAG-TwinStrep sequence in AAVS1-Puro-PGK1−3 × FLAG TwinStrep was replaced with an HA-tagged SLC37A3 CDS. |
| Recombinant DNA reagent | AAVS1-Puro-PGK1-s/lATRAID-V5 | this paper | | The FLAG-TwinStrep sequence in AAVS1-Puro-PGK1−3 × FLAG TwinStrep was replaced with an V5-tagged short/long ATRAID CDS. |
| Recombinant DNA reagent | pLenti PGK Hygro s/lATRAID-V5 | this paper | | A V5-tagged short/long ATRAID CDS was inserted under the PGK promoter for lentiviral expression |
| Sequence-based reagent | sgRNA_human_ATRAID_exon3_1 | this paper | sequence: GCCTGAT GAAAGTTTGGACC | a sgRNA targeting exon 3 of ATRAID for generating knock-out cells |
| Sequence-based reagent | sgRNA_human_ATRAID_exon3_2 | this paper | sequence: CCCTGGTC CAAACTTTCATC | a sgRNA targeting exon 3 of ATRAID for generating knock-out cells |
| Sequence-based reagent | sgRNA_human_ATRAID_exon5 | this paper | sequence: GTCCTGGA GGAATTAATGCC | a sgRNA targeting exon 5 of ATRAID for generating knock-out cells |
| Sequence-based reagent | sgRNA_human_SLC37A3_intron5_1 | this paper | sequence: GTGTGAGTG TATCCTTCACG | a sgRNA targeting intron 5 of SLC37A3 for generating knock-out cells |
| Sequence-based reagent | sgRNA_human_SLC37A3_intron5_2 | this paper | sequence: GCCAGTGCCT GTAAGTCACG | a sgRNA targeting intron 5 of SLC37A3 for generating knock-out cells |
| Sequence-based reagent | sgRNA_human_SLC37A3_intron6 | this paper | sequence: GTAGCAAGTC AGAGTTGTTCA | a sgRNA targeting intron 6 of SLC37A3 for generating knock-out cells |
| Sequence-based reagent | sgRNA_mouse_SLC37A3_exon1_1 | this paper | sequence: TCTCTGCAA AAATCGTGGCC | a sgRNA targeting exon 2 of SLC37A3 for generating knock-out cells |
| Sequence-based reagent | sgRNA_mouse_SLC37A3_exon1_2 | this paper | sequence: TGTTCCTGC TCACGTTCTTC | a sgRNA targeting exon 2 of SLC37A3 for generating knock-out cells |
| Peptide, recombinant protein | HA peptide | ThermoFisher | ThermoFisher: 26184 | |
| Peptide, recombinant protein | V5 peptide | APExBIO | APExBIO: A6005 | |
| Commercial assay or kit | CellTiter-Glo | Promega | Promega: G7572 | |
| Commercial assay or kit | anti-HA affinity matrix | Sigma-Aldrich | Sigma-Aldrich: A2095 | |
| Commercial assay or kit | anti-V5 affinity matrix | Sigma-Aldrich | Sigma-Aldrich: A7345 | |
| Chemical compound, drug | alendronate | Sigma-Aldrich | Sigma-Aldrich: A4978 | |
| Chemical compound, drug | zoledronate | Sigma-Aldrich | Sigma-Aldrich: SML0223 | |
| Chemical compound, drug | ibandronate | Sigma-Aldrich | Sigma-Aldrich: I5784 | |

*Continued*

| Reagent type (species) or resource | Designation | Source or reference | Identifiers | Additional information |
|---|---|---|---|---|
| Chemical compound, drug | lovastatin | Sigma-Aldrich | Sigma-Aldrich: PHR1285 | |
| Chemical compound, drug | AlexaFlour 647 labeled zoledronate | BioVinc | BioVinc: AF647-ZOL | |
| Chemical compound, drug | digitonin | Millipore-Sigma | Millipore-Sigma: 300410 | |
| Chemical compound, drug | saponin | Sigma-Aldrich | Sigma-Aldrich: 47036 | |
| Software, algorithm | Huygens Professional | Scientific Volume Imaging | RRID:SCR_014237 | |

## Materials

Reagents were obtained from the following sources: antibodies against LAMP2 (mouse, sc-18822), Ran BP3 (mouse, sc-373678) and Rap 1A (goat, sc-1482) were from Santa Cruz Biotechnology (Dallas, Texas); antibodies against GAPDH (rabbit, 2118), V5-tag (rabbit, 13202) and EEA1 (rabbit, 2411) were from Cell Signaling Technology (Danvers, Massachusetts); antibodies against $Na^+/K^+$-ATPase (mouse, ab7671) and Lamin B1 (rabbit, ab16048) were from Abcam (Cambridge, Massachusetts); antibodies against V5-tag (mouse, R960-25) and HDJ2 (mouse, MS-225-P0) were from ThermoFisher Scientific (Waltham, Massachusetts); antibodies against Caveolin-1 (rabbit, C3237) and HA-tag (rat, 11867423001) were from Sigma-Aldrich (Burlington, Massachusetts); antibody against EEA1 (mouse, 610456) was from BD Biosciences (San Jose, California); antibody against LMAP1 (mouse, H4A3) was from DSYB (Developmental Studies Hybridoma Bank, Iowa City, Iowa); antibody against Ubiquitin (mouse, 05–944) was from Millipore-Sigma (Burlington, Massachusetts); alendronate (A4978), lovastatin (PHR1285), zoledronate (SML0223), ibandronate (I5784), etidronate (P5248), tiludronate (T4580), chloroquine (C6628), puromycin (P8833), polybrene (H9268), anti-HA (A2095) and anti-V5 (A7345) agarose affinity matrix, poly-L-lysine solution (P4707), fibronectin solution (F0895), Triton X-100 (T8787), saponin (47036), Bovine Serum Albumin (BSA, A9647) and complete protease inhibitor cocktail (11836170001) were from Sigma-Aldrich; HRP-conjugated anti-rat secondary antibody (31470), Alexa 488 and Alexa 647-conjugated secondary antibodies (A21208, A32728, A11034 and A32733), 0.1 μm TetraSpeck microspheres (T7279), SlowFade Diamond mounting medium (S36968), Halt protease-phosphatase inhibitor cocktail (78443), BCA protein assay kit (23225), SuperSignal west femto substrate (34095), TOPO TA cloning kit (450030), TRIzol and TRIzol LS reagents (15596018 and 10296028), SuperScript IV (18090050), RNase-free Turbo DNase (AM2238), SUPERase. In RNase Inhibitor (AM2694), SYBR Green qPCR master mix (A25742), Hygromycin B (10687010), DMEM (11965118), RPMI (11875093), Fetal Bovine Serum (FBS, 16000044) and Lipofectamine 3000 reagent (L3000008) were from ThermoFisher Scientific; HRP-conjugated anti-mouse and anti-rabbit secondary antibodies (1706515 and 1706516) were from Bio-Rad (Hercules, California); PNGase F (P0704) and Endo $H_f$ (P0703) were from New England Biolabs (NEB, Ipswich, Massachusetts); cell line Nucleofector kit V (VACA-1003) was from Lonza (Walkersville, Maryland); CellTiter-Glo kit (G7572) was from Promega (Fitchburg, Wisconsin); mouse RANK ligand (RANKL, 462-TEC-010) was from R and D systems (Minneapolis, Minnesota); IMDM (30–2005) was from ATCC (American Type Culture Collection, Manassas, Virginia); digitonin (300410) was from Millipore-Sigma; QuickExtract DNA extraction solution (QE0905T) was from Epicentre (Madison, Wisconsin); tritium-labeled alendronate (MT-1727) was from Moravek (Brea, California); Alexa flour 647 labeled zoledronate (AF647-ZOL) was from BioVinc (Pasadena, California); Janelia flour 549 was a gift from Luke Lavis; pSpCas9(BB)−2A-GFP (PX458) was a gift from Feng Zhang (Addgene plasmid # 48138); AAVS1-Puro-PGK1 −3 × FLAG TwinStrep was a gift from Yannick Doyon (Addgene plasmid # 68375); pLenti PGK Hygro DEST (w530-1) was a gift from Eric Campeau and Paul Kaufman (Addgene plasmid # 19066); psPAX2 and pMD2.G were gifts from Didier Trono (Addgene plasmid # 12260 and # 12259).

## Cell lines and tissue culture

K562 human myeloid leukemia cells, Human Embryo Kidney (HEK) 293 T cells and RAW 264.7 murine macrophage-like cells were obtained from ATCC. Cell line identities were verified by determining species identity and examining morphology. All cell lines were tested for mycoplasma contamination using ATCC universal mycoplasma detection kit and/or DAPI staining. All cell lines were free of mycoplasma contamination. K562 cells were cultured in RPMI supplemented with 25 mM HEPES, 2 mM L-glutamine, 2 g/L NaHCO$_3$, 10% FBS and penicillin/streptomycin; HEK 293 T cells and RAW 264.7 cells were cultured in DMEM supplemented with 10% FBS and penicillin/streptomycin. All cultures were maintained at 37°C and 5% CO$_2$.

## Unbiased whole-genome CRISPRi screen

K562 cell line generation, genome-scale library design and cloning, virus production, and bioinformatic analysis were conducted as previously described (*Gilbert et al., 2014*; *Jost et al., 2017*). In summary, K562 cells stably expressing dCas9-KRAB were transduced in duplicate with the v1 CRISPRi sgRNA library to achieve ~30% infection to ensure no more than one viral integration event per cell. Two days after transduction, cells were selected with 0.75 µg/mL of puromycin for 2 days and then kept with fresh puromycin-free medium for 2 days for recovery. At this point (t$_0$), 250 million cells (ensuring a minimum of 1000 × library coverage) were harvested from each replicate and the remaining cells in each replicate were split into two populations for untreated growth and alendronate-treated growth. For alendronate treatment, cells were cultured in medium containing 250 µM alendronate for 24 hr, spun down to remove the drug and re-suspended in fresh medium. Cells were cultured for another 13 days to allow the untreated population to double seven more times than the alendronate-treated population. 250 million cells were then harvest from each group (two replicates for each condition, four groups in total). Cells were maintained at a density of 500,000 to 1,000,000 cells/mL in 2-liter cultures to ensure a library coverage of at least 1000 cells per sgRNA during the entire screening period. Genomic DNA was collected from all harvested samples and the genomic regions containing the inserted sgRNAs were amplified for 20 cycles by PCR and sequenced at 800 × coverage on Ilumina HiSeq-2500 using custom primers as previously described (*Kampmann et al., 2013*).

For data analysis, sequencing reads were aligned to the v1 CRISPRi library sequences, counted, and quantified using the Python-based ScreenProcessing pipeline (*Horlbeck et al., 2016*). Sensitivity phenotypes (ρ) were calculated by computing the log2 difference in enrichment of each sgRNA between the treated and untreated samples, subtracting the equivalent median value for all non-targeting sgRNAs, and dividing by the number of population doubling differences between the treated and untreated populations (*Gilbert et al., 2014*; *Jost et al., 2017*). Similarly, untreated growth phenotypes (γ) were calculated from the untreated endpoint samples and t$_0$ samples, dividing by the total number of doublings of the untreated population. Phenotypes from sgRNAs targeting the same gene were collapsed into a single sensitivity phenotype for each gene using the average of the top three scoring sgRNAs (by absolute value) and assigned a *P*-value using the Mann-Whitney test of all sgRNAs targeting the same gene compared to the non-targeting controls (*Supplementary file 1*). For genes with multiple independent transcription start sites (TSSs) targeted by the sgRNA libraries, phenotypes and *P*-values were calculated independently for each TSS and then collapsed to a single score by selecting the TSS with the lowest Mann-Whitney *P*-value. Replicate-averaged sensitivity phenotype and *P*-value for each gene were obtained by performing the above computations on the average of sgRNA phenotype values calculated from both replicates and used for illustration. The CRISPRi screen was performed only once.

## Generation of knockout cell lines in K562, HEK 293T and RAW cells

Genome editing experiments were designed based on an established protocol (*Ran et al., 2013*). For the human *ATRAID* locus, one sgRNA targeting exon three and another targeting exon five were used to act simultaneously and remove part of exon 3, the entire exon four and part of exon 5. For the human *SLC37A3* locus, one sgRNA targeting intron five and another targeting intron six were used to act simultaneously and remove the entire exon 6, which contains 146 bp of CDS. For murine *SLC37A3* locus, two sgRNAs targeting exon two were designed to cause microdeletions and frameshifts in the CDS. sgRNAs were cloned into PX458 for co-expression with Cas9.

sgRNA_human_*ATRAID*_exon3_1: GCCTGATGAAAGTTTGGACC
sgRNA_human_*ATRAID*_exon3_2: CCCTGGTCCAAACTTTCATC
sgRNA_human_*ATRAID*_exon5: GTCCTGGAGGAATTAATGCC
sgRNA_human_*SLC37A3*_intron5_1: GTGTGAGTGTATCCTTCACG
sgRNA_human_*SLC37A3*_intron5_2: GCCAGTGCCTGTAAGTCACG
sgRNA_human_*SLC37A3*_intron6: GTAGCAAGTCAGAGTTGTTCA
sgRNA_mouse_ *SLC37A3*_exon2_1: TCTCTGCAAAAATCGTGGCC
sgRNA_mouse_ *SLC37A3*_exon2_2: TGTTCCTGCTCACGTTCTTC

For K562 and HEK 293 T cells, on day one, 500,000 cells were seeded onto a 6 cm dish. 24 hr later, cells were transfected with 1.25 µg of PX458 construct for each sgRNA (total of 2.5 µg DNA) using the Lipofectamine 3000 reagent according to manufacturer instructions. Medium was replenished on the following day. 48 hr after transfection, cells were trypsinized, filtered through a 50 µm strainer into ice-cold FACS buffer (PBS containing 1% FBS) and sorted with a flow cytometer for GFP-positive cells. Single GFP-positive cells were seeded into the wells of a 96-well plate containing 150 µL of DMEM in each well. 12 to 14 days later, each surviving clone was split into two wells, with one well saved for expansion and the other for genotyping. For genotyping, genomic DNA was extracted from each confluent clone with QuickExtract solution and used to perform genomic PCR using a pair of primers flanking the target region. Successful deletion events were identified by a significant decrease in the size of PCR products. Clones with deletions on all alleles of the target gene were further expanded and stored. Genomic PCR products from clones with homozygous deletions were inserted into TOPO-TA cloning vectors and sequenced to identify clones that have frameshifts on all alleles of the target gene. $ATRAID^{KO}$; $SLC37A3^{KO}$ cells are generated by knocking out *ATRAID* in $SLC37A3^{KO}$ cells.

human_*ATRAID*_KO_verification_forward: CTGAAAAGGGGGGTTGTGTAGTCAA
human_*ATRAID*_KO_verification_reverse: GGGTTATAGCCCCAGAACTCTGAA
human_*SLC37A3*_KO_verification_forward: GTTGGAGGGCTGATAGCTTAATG
human_*SLC37A3*_KO_verification_reverse: AAAAATTGAGACCTCCTGCCTTG

For RAW 264.7 cells, 2 million cells were electroporated with 2 µg of PX458 harboring the sgRNAs of interest (1 µg per sgRNA construct) in 100 µL of Nucleofector solution V on a Nucleofector device (Lonza) using program D-023. Cells were allowed to recover for two days before they were single-cell sorted into a 96-well plate. Clonal expansion and genotyping were then performed as described above for HEK 293 T cells.

Mouse_*SLC37A3*_KO_verification_forward: CCCACAGGCAGAAGACAAGA
Mouse_ *SLC37A3*_KO_verification_reverse: TGTAACTCAGTCACTGGGAGGA

## Differentiation of RAW macrophages into osteoclasts

Differentiation of RAW cells to osteoclasts was achieved following an established protocol (*Collin-Osdoby and Osdoby, 2012*). Briefly, RAW 264.7 cells were seeded into a 24-well plate and treated with 35 ng/mL RANKL for 6 days to reach a large, multi-nucleated morphology that is characteristic of osteoclasts. For experiments with alendronate, the drug was added at the indicated concentrations 48 hr prior to harvesting. RAW cell differentiation was repeated independently for three times.

## Cell viability assays

On day one, cells were seeded at 8000 cells per well for K562 and HEK 293 T cells or 4000 cells per well for RAW 264.7 cells in a 96-well plate and treated with a series of doses of the desired drug. Three wells were prepared for each combination of cell line and drug concentration. Forty-eight hours later, the total cellular ATP level in each well was measured using the CellTiter-Glo luminescent assay following manufacturer instructions. Relative ATP levels were then plotted as percentages of ATP levels in the untreated samples and interpreted as a proxy for cell viability under drug treatments. Each viability curve was repeated independently for at least two times.

## Generation of HEK 293 T cells stably expression epitope tagged ATRAID and SLC37A3

V5-tagged ATRAID and HA-tagged SLC37A3 were constructed by appending codon-optimized sequences of V5 (sequence: GGA AAG CCC ATA CCG AAT CCT CTC CTT GGG TTG GAT AGC

ACT) and HA tags (sequence: TAC CCC TAT GAT GTT CCT GAT TAC GCG) to the C-termini of *ATRAID* (both the short isoform, 229 a.a., and the long isoform, 284 a.a.) and *SLC37A3* CDSs, respectively. A GGGGSGGGGS flexible linker (sequence: GGT GGA GGG GGA AGT GGC GGA GGA GGT TCA) was added between each CDS and its epitope tag.

To generate HEK 293 T cells that stably express sATRAID-V5, lATRAID-V5 or SLC37A3-HA at sub-endogenous levels, CDSs were cloned into pPGK-AAVS-Puro (derived from Addgene # 68375) and inserted into human *AAVS1* locus via CRISPR-Cas9 mediated homologous recombination. For genome integration, a 40% confluent dish of $ATRAID^{KO}$, $SLC37A3^{KO}$ or $KO^2$ HEK 293 T cells was transfected with 1.5 µg of pPGK-AAVS-Puro carrying the desired CDS and 1 µg of PX458 expressing sgRNA_*AAVS1* (see below) using Lipofectamine 3000 reagent. Culture medium was replenished on the next day. 48 hr after transfection, cells were re-plated in a 10 cm dish in DMEM containing 2 µg/mL puromycin to select for successfully edited cells. Puromycin selection was continued for a week, with medium replenished every other day. After selection, expression levels of epitope-tagged proteins were analyzed by immunoblotting and RT-qPCR as described below.

sgRNA_*AAVS1*: GGGGCCACTAGGGACAGGAT

To generate HKE 293 T cells that co-express ATRAID-V5 and SLC37A3-HA at near-endogenous levels, the CDS of sATRAID-V5 or lATRAID-V5 was cloned into w530-1 and transduced into $KO^2$ +-SLC37A3-HA cells. To generate the lentiviruses, on day one 150,000 HEK 293 T cells were seeded into a well of a 6-well plate. 24 hr later, cells were transfected with 2 µg w530-1 construct, 0.8 µg psPAX2 and 0.4 µg pMD2.G using Lipofectamine 3000 reagent. On the next day culture medium was replenished. 48 hr after transfection, supernatant from the culture was collected and filtered through a 0.45 µm filter. 150 µL of the viral medium was added to a 40% confluent well of $KO^2$ +-SLC37A3-HA cells in a 6-well plate in the presence of 8 µg/mL Polybrene to a total volume of 2 mL. 48 hr later cells were re-plated into a 6 cm dish in DMEM containing 200 µg/mL hygromycin. Selection was continued for one week, with medium replenished every other day. After selection, expression levels of epitope-tagged proteins were analyzed by immunoblotting and RT-qPCR as described below.

## Immunoblot assays

Cells were washed once with PBS and lysed on ice by scraping into ice-cold RIPA buffer supplemented with Halt protease-phosphatase inhibitor. The lysate was then cleared by centrifuging at 20,000 × g, 4°C for 15 min. Protein concentration in the lysate was determined with BCA protein assays. Loading samples were prepared by mixing lysates with SDS loading buffer and incubating at 37°C for 15 min. (Higher denaturing temperatures may cause SLC37A3 to aggregate and prevent it from entering the gel.) SDS-PAGE electrophoresis and protein transfer onto nitrocellulose membranes were performed according to standard protocols. Membranes were blocked in TBST containing 2.5% BSA and 2.5% skim milk for 1 hr and incubated with primary antibodies overnight in TBST containing 5% BSA. (Primary antibody concentrations: α-HA, 100 ng/mL; α-GAPDH and α-HDJ-2, 250 ng/mL; α-V5 (mouse), α-LAMP1, and α-Lamin B1, 500 ng/mL; α-Rap 1A, α-Ran BP3, α-EEA1 (rabbit) and α-Caveolin-1, 1 µg/mL.) Membranes were then washed 3 × 5 min in TBST and incubated with either HRP-conjugated or fluorophore-conjugated secondary antibodies (1:2000 dilution for all secondary antibodies) in TBST containing 5% skim milk for 1 hr. Membranes were then washed again for 3 × 5 min in TBST and visualized with either SuperSignal substrate or a Typhoon scanner. Each blot was repeated independently for two times.

## RT-qPCR

RNA was extracted from near-confluent 3 cm dishes using TRIzol reagent following manufacturer instructions. Purified RNA was reverse transcribed using SuperScript IV and oligo d(T)$_{20}$ following manufacturer instructions. qPCR reactions were performed with SYBR Green qPCR master mix and primers listed below using a CFX96 or CFX384 Real-Time PCR machine (Bio-Rad). Ct values were calculated for each transcript using triplicate measurements, and relative mRNA levels were determined for each gene using *TBP* (TATA binding protein) and *RPLP1* (60S acidic ribosomal protein P1) as loading references for human transcripts and *Actb* (β actin) and *Rplp0* (60S acidic ribosomal protein P0) as loading references for mouse transcripts. Each qPCR measurement was repeated independently for two times.

human_*TBP*_forward: ATAAGAGAGCCACGAACCACG
human_*TBP*_reverse: TGCCAGTCTGGACTGTTCTTC
human_*RPLP1*_forward: AGCCTCATCTGCAATGTAGGG
human_ *RPLP1*_reverse: TCAGACTCCTCGGATTCTTCTTT
human_*ATRAID*_forward: CAGAAGGGCACCATCTTGGG
human_*ATRAID*_reverse: ACCTTTGAGGGGGTTTGCTT
human_*SLC37A3*_forward: GCTGCCTGTGGATTGTGAAC
human_*SLC37A3*_reverse: AAAATGTTGCCCACCGAAGC
mouse_*Actb*_forward: TGTCGAGTCGCGTCCA
mouse_*Actb*_reverse: ATGCCGGAGCCGTTGTC
mouse_*Rplp0*_forward: TGCTCGACATCACAGAGCAG
mouse_*Rplp0*_reverse: ACGCGCTTGTACCCATTGAT
mouse_*Ctsk*_forward: CCTTCCAATACGTGCAGCAG
mouse_*Ctsk*_reverse: CATTTAGCTGCCTTTGCCGT
mouse_*Rank*_forward: GCAGCTCAACAAGGATACGG
mouse_*Rank*_reverse: TAGCTTTCCAAGGAGGGTGC
mouse_*Trap*_forward: AAGAGATCGCCAGAACCGTG
mouse_*Trap*_reverse: CGTCCTCAAAGGTCTCCTGG

## Immunofluorescence assays

On day one, coverslips are placed into wells of 6-well plates and coated for 1 hr at 37°C with 0.01% Poly-L-lysine solution supplemented with 10 µg/mL fibronectin. Coating solution was aspirated and 120,000 HEK 293 T cells were plated into each well. Twenty-four hours later, the coverslips were rinsed two times with PBS$^{++}$ (PBS containing calcium and magnesium) and fixed in 4% formaldehyde in PBS$^{++}$ for 15 min at room temperature. The coverslips were rinsed three times with PBS$^{++}$ and cells were permeabilized/blocked with 0.1% Saponin and 2% BSA in PBS$^{++}$ for 30 min. After rinsing briefly with PBS$^{++}$, the coverslips were transferred to a humidity chamber and incubated overnight at 4°C in PBS$^{++}$ containing 2% BSA and desired primary antibodies. (Antibody concentrations: α-HA and α-EEA1 (mouse), 500 ng/mL; α-LAMP2, 1 µg/mL; α-V5 (rabbit), 2 µg/mL; α-Na$^+$/K$^+$-ATPase, 5 µg/mL.) On the next day, the coverslips were washed 3 × 5 min in PBS$^{++}$ and incubated for 1 hr with Alexa-488 and Alexa-647-conjugated secondary antibodies diluted in PBS$^{++}$ containing 2% BSA. The coverslips were again washed 4 × 5 min in PBS$^{++}$ and mounted onto slides in SlowFade Diamond anti-fade mountant supplemented with DAPI, and the edges of the coverslips were sealed with nail polish. (We do not recommend curing mountants such as ProLong Gold, as they distort/flatten the samples. We also do not recommend Vectashield if Alexa 647 is chosen as a fluorophore.)

Images were acquired on a Zeiss AxiObserver Z.1 microscope equipped with a Zeiss α Plan-APO 100× (NA1.46) oil-immersion objective, a 561 nm Hamamatsu Gemini 2C beam splitter, dual Hamamatsu Image-EM 1K EM-CCD cameras and CoolLED P4000 light sources. The ZEN blue software package (Zeiss) was used to control the hardware and image acquisition conditions. An LED-DA/FI/TR/Cy5-A quadruple band-pass filter set was installed in the microscope turret and used for all channels. Excitation wavelength (EW), excitation filters (ExF) and emission filters (EmF) used for each channel are: DAPI channel, EW 385 nm, ExF 387/11 nm, EmF 525/45 nm; Alexa 488 channel, EW 470 nm, ExF 473/10 nm, EmF 525/45 nm; Alexa 647 channel, EW 635 nm, ExF 635/6 nm, EmF 682/40 nm. Images were sampled above the Nyquist limit with a voxel size of 81.25 nm ×81.25 nm×220 nm. Each IF sample was independently prepared and imaged for two times.

With each batch of samples, one slide of 0.1 µm TetraSpeck microspheres immersed in SlowFade Diamond was also prepared and imaged with the same settings. The point-spread function (PSF) of each channel was distilled from the microsphere images and used to deconvolute sample images using Huygens Professional deconvolution software (Scientific Volume Imaging). Chromatic aberration (shift, rotation and scaling) were estimated by correlating different channels of the microsphere images. The estimated aberration parameters were then used to align different channels of deconvoluted images using Huygens Profession software. Only images that have been deconvoluted and aligned were used for analysis. Images displayed in figures are representative single Z-slices.

## Live cell imaging

150,000 HEK 293 T cells expressing SLC37A3-Halo were seeded into a 35 mm glass bottom tissue culture dish (MatTek) coated with poly-L-lysine and fibronectin (see the previous section). Cells were cultured in DMEM containing 500 nM AF647-ZLN for 16 hr. Cells were washed once in PBS and stained with DMEM containing 100 nM JF549 for 30 min, washed three times with PBS, incubated in fresh DMEM for 30 min, and washed again in PBS for three times and finally cultured in DMEM without phenol Red to facilitate live cell imaging. The glass bottom dish was then placed in an incubation chamber mounted onto a microscope, and images were taken and analyzed as described in the previous section.

## Reciprocal co-immunoprecipitation

10 cm dishes of near-confluent $KO^2$ HEK 293 T cells over-expressing the proteins indicated in *Figure 3Ff* were lysed by scraping on ice into 1 mL of ice-cold lysis buffer (1% Trition X-100, 20 mM Tris-HCl pH 8.0, 150 mM NaCl and 2 mM EDTA in $ddH_2O$) supplemented with cOmplete protease inhibitors. The lysates were cleared as described above. 10 µL of each cleared lysate was saved for input analysis. The rest of the lysates were transferred to Eppendorf tubes containing 20 µL (settled volume) of anti-HA or anti-V5 agarose beads that had been blocked overnight in lysis buffer containing 2% BSA. Lysates were incubated with the beads for 90 min at 4°C, washed three times in low-salt wash buffer (0.1% Triton X-100, 10 mM Tris-HCl pH 7.5, 150 mM NaCl and 1 mM EDTA in $ddH_2O$) supplemented with protease inhibitors and three times in high-salt wash buffer (0.1% Triton X-100, 10 mM Tris-HCl pH 7.5, 300 mM NaCl and 1 mM EDTA in $ddH_2O$) supplemented with protease inhibitors, and eluted in 100 µL elution buffer (1% Triton X-100, 10 mM Tris-HCl pH 7,5 and 150 mM NaCl) containing 2 mg/mL HA or V5 peptide. Inputs and eluates were then analyzed by immunoblotting as described. The co-IP experiment was repeated independently for two times.

## Analysis of protein glycosylation

Protein lysates and IP eluates were prepared as described above. Once obtained, samples (lysates containing 20–40 µg of total protein or IP eluates that correspond to 60 µg of total protein input) were either left untreated, or treated with peptide-N-Glycosidase F (PNGase F) or endo-glycosidase H (Endo H) following manufacturer instructions (NEB) using non-denaturing conditions (without adding SDS or boiling for denaturation). The reactions were incubated at 37°C overnight and subsequently analyzed by SDS-PAGE electrophoresis and immunboblotting.

## Inferring translation efficiency with polysome profiling

For each cell line of interest, one near-confluent 15 cm dish of cells was lysed on ice by scraping into 300 µL of ice-cold lysis buffer (1% Triton X-100, 20 mM Tris-HCl pH 7.0, 20 mM Tris-HCl pH 8.0, 15 mM $MgCl_2$, 150 mM NaCl, 5 mM $CaCl_2$, 25 U/mL RNase-free Turbo DNase, 500 U/mL SUPERase.In RNase Inhibitor and 0.1 mg/mL cycloheximide in RNase-free water) supplemented with protease inhibitors. Lysates were homogenized by passing through a 22-gauge needle for 10 times and then cleared as described above. 10–50% Sucrose gradients were composed by mixing 6 mL of 10% Sucrose solution (20 mM Tris-HCl pH 7.0, 20 mM Tris-HCl pH 8.0, 15 mM $MgCl_2$, 150 mM NaCl, 0.1 mg/mL cycloheximide and 10% (w/v) sucrose in RNase-free water) with 6 mL of 50% sucrose solution (20 mM Tris-HCl pH 7.0, 20 mM Tris-HCl pH 8.0, 15 mM $MgCl_2$, 150 mM NaCl, 0.1 mg/mL cycloheximide and 50% (w/v) sucrose in RNase-free water) on a Gradient Master device (BioComp). 450 µL of each cleared lysate was loaded onto a 10–50% sucrose gradient and centrifuged at 35,000 rpm, 4°C for 2.5 hr using a SW 41 Ti rotor (Beckman). At the same time, 50 µL of each cleared lysate was used to extract total RNA. The density-separated RNA samples were loaded onto a Gradient Master device, collected from the top (low density) to the bottom (high density) and analyzed by measuring $A_{254}$. Fractionation of samples was carried out manually by visually determining the boundaries between desired fractions during sample analysis. RNA was then extracted from each fraction using TRIzol LS reagent following manufacturer instructions. Levels of *SLC37A3* transcripts in collected fractions were then measured by RT-qPCR as described above and the distribution of *SLC37A3* transcript in the differentially translated fractions was plotted and interpreted as a proxy for translation efficiency. The polysome profiling experiment was repeated independently for two times.

## Radioactive uptake assays

For whole-cell uptake assays, on day one 1.5 million WT, ATRAID$^{KO}$ or SLC37A3$^{KO}$ HEK 293 T cells were seeded into each poly-L-lysine coated 6 cm dish in DMEM containing 1 µCi/mL $^3$H-alendronate. 24 hr later, cells were washed three times with PBS$^{++}$, each time thoroughly removing the PBS. Cells were then trypsinized, pelleted and re-suspended in PBS. Cell density in each sample was measured on a Multisizer 3 Coulter Counter (Beckman Coulter). Total intracellular radioactivity in 2 million cells from each sample was measured by scintillation counting on a LS6500 Liquid Scintillation Counter (Beckman Coulter). Three plates were used as triplicates for each cell line.

Fractionation-based uptake assays were designed based on an established protocol (*Liu and Fagotto, 2011*). On day one 6 million WT, ATRAID$^{KO}$ or SLC37A3$^{KO}$ HEK 293 T cells were seeded into each Poly-L-lysine coated 10 cm dish in DMEM containing 1 µCi/mL $^3$H-alendronate. 24 hr later, dishes were transferred to a cold room and washed three times with 5 mL of PBS$^{++}$, each time thoroughly removing the PBS. 3 mL of permeabilization buffer (20 mM HEPES pH 7.4, 150 mM NaCl, 0.2 mM EDTA, 2 mM MgCl$_2$, 2 mM DTT and 42 µg/mL digitonin in ddH$_2$O) supplemented with protease inhibitors was then added to each dish, and the dishes were incubated on an orbital shaker (100 rpm) at 4°C for 10 min. 2.5 mL supernatant was collected from each dish as the cytosolic fraction, the rest of the supernatant was thoroughly aspirated. The permeabilized cells were then washed again in 5 mL of PBS$^{++}$ and lysed in 500 µL of RIPA buffer supplemented with protease inhibitors. The RIPA lysates were cleared as described above and collected as membranous fractions. Radioactivity in cytosolic and membranous fractions was measured by scintillation counting and normalized to protein concentrations measured by BCA protein assays. Two plates were used as duplicates for each cell line.

Lysosome purification-based uptake assays were adapted from established protocols (*Abu-Remaileh et al., 2017*; *Wyant et al., 2017*). Three 15 cm dishes of ~35 million WT, ATRAID$^{KO}$ or SLC37A3$^{KO}$ HEK 293 T cells expressing Tmem192$-3 \times$ HA were used for each experiment. Cells were incubated in RPMI containing 1 µCi/mL $^3$H-alendronate and incubated for 3 hr, quickly rinsed twice with PBS, scraped in 1 mL of KPBS (136 mM KCl, 10 mM KH$_2$PO$_4$, pH 7.25 was adjusted with KOH), pelleted by centrifuging at 1000 × g, 4°C for 2 min and re-suspended in 950 µL KPBS. The cell suspension was then homogenized with 20 strokes of a 2 mL dounce homogenizer and centrifuged at 1000 × g, 4°C for 2 min. The supernatant was then incubated with 150 µL of KPBS-prewashed anti-HA magnetic beads for 3 min. The beads were then gently washed three times with KPBS and then re-suspended in 50 µL ice-chilled extraction buffer (80% methanol and 20% ddH$_2$O) and incubated for 5 min on ice. The lysosome extract was then centrifuged at 1000 × g, 4°C for 2 min. Radioactivity in the supernatant was then measured by scintillation counting.

All uptake assays were repeated independently for two times.

## Statistical tests

*P*-values assigned to individual genes in the CRISPRi screen were calculated using non-parametric Mann-Whitney *U* test (11, 23). *P*-values in *Figure 4b* were calculated using unpaired two-way ANOVA test. *P*-values in *Figure 4c* were calculated using two-tailed unpaired t-test assuming equal variance. Degrees of freedom, *F* values and t values are reported in figure legends.

## Data availability

The data that support the findings in this study are available within the paper and its supplementary files.

## Acknowledgement

We thank Dr. A Gutu and Dr. J Wisniewski for assistance with microscope setup and imaging. We thank Dr. A Darnell for assistance with polysome profiling experiments. We thank members of the O'Shea lab for insightful discussions. This work was supported by the Howard Hughes Medical Institute.

## Additional information

### Competing interests

Erin K O'Shea: Chief Scientific Officer and a Vice President at the Howard Hughes Medical Institute, one of the three founding funders of *eLife*. The other authors declare that no competing interests exist.

### Funding

| Funder | Author |
|---|---|
| Howard Hughes Medical Institute | Zhou Yu<br>Lauren E Surface<br>Chong Yon Park<br>Max A Horlbeck<br>Gregory A Wyant<br>Monther Abu-Remaileh<br>David M Sabatini<br>Jonathan S Weissman<br>Erin K O'Shea |

The funders had no role in study design, data collection and interpretation, or the decision to submit the work for publication.

### Author contributions

Zhou Yu, Conceptualization, Resources, Data curation, Formal analysis, Validation, Investigation, Visualization, Methodology, Writing—original draft; Lauren E Surface, Conceptualization, Data curation, Formal analysis, Validation, Investigation; Chong Yon Park, Resources, Investigation, Methodology; Max A Horlbeck, Resources, Data curation, Software, Formal analysis, Methodology; Gregory A Wyant, Conceptualization, Resources, Formal analysis, Investigation, Methodology; Monther Abu-Remaileh, Conceptualization, Resources, Investigation, Methodology; Timothy R Peterson, Conceptualization, Data curation, Formal analysis, Project administration; David M Sabatini, Jonathan S Weissman, Conceptualization, Resources, Supervision, Project administration; Erin K O'Shea, Conceptualization, Resources, Supervision, Funding acquisition, Validation, Writing—original draft, Project administration

### Author ORCIDs

Zhou Yu http://orcid.org/0000-0003-3213-4583
Max A Horlbeck http://orcid.org/0000-0002-3875-871X
Jonathan S Weissman http://orcid.org/0000-0003-2445-670X
Erin K O'Shea http://orcid.org/0000-0002-2649-1018

### Decision letter and Author response

Decision letter https://doi.org/10.7554/eLife.36620.019
Author response https://doi.org/10.7554/eLife.36620.020

## Additional files

### Supplementary files

• Supplementary file 1. A data table containing statistical information on all genes targeted in the CRISPRi screen using alendronate as the selection agent. Information on three phenotypic scores, gamma (impact of CRISPRi silencing on basal cell growth), rho (impact of CRISPRi silencing on alendronate-mediated inhibition of cell growth) and tau (combined impact of CRISPRi silencing on overall cell growth in the presence of alendronate), are reported for all genes targeted in the screen in each replicate of the screen (RepA and RepB) as well as the average of the two replicates (ave_RepB_RepA). Specifically, numeric values and statistical significance of the three phenotypic scores are listed for all genes targeted in the screen.
DOI: https://doi.org/10.7554/eLife.36620.013

• Supplementary file 2. A data table containing statistical information on all genes targeted in the CRISPRi screen using zoledronate as the selection agent. Statistical information reported is the same as in *Supplementary file 1*.
DOI: https://doi.org/10.7554/eLife.36620.014

• Transparent reporting form
DOI: https://doi.org/10.7554/eLife.36620.015

### Data availability

All data generated or analysed during this study are included in the manuscript and supporting files. Source data files have been provided for the two CRISPRi screens shown in Figure 1 and its figure supplement.

The following dataset was generated:

| Author(s) | Year | Dataset title | Dataset URL | Database, license, and accessibility information |
|---|---|---|---|---|
| O'Shea EK, Yu Z, Surface LE, Park CY, Horlbeck M, Wyant G, Abu-Remaileh M, Peterson TR, Sabatini DM, Weissman JS | 2018 | Data from: Identification and characterization of a transporter complex responsible for the cytosolic entry of nitrogen-containing-bisphosphonates | https://doi.org/10.5061/dryad.p6261d6 | Available at Dryad Digital Repository under a CC0 Public Domain Dedication |

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
