## [Decision Letter]

Thank you for submitting your article "Identification of a transporter complex responsible for the cytosolic entry of nitrogen-containing-bisphosphonates" for consideration by *eLife*. Your article has been reviewed by three peer reviewers, and the evaluation has been overseen by Philip Cole as the Senior/Reviewing Editor. The following individual involved in review of your submission has agreed to reveal his identity: Bruno Gasnier (Reviewer #1).

The reviewers have discussed the reviews with one another and the Reviewing Editor has drafted this decision to help you prepare a revised submission.

Summary:

In this study, the authors performed a genome-wide CRISPR interference screen to identify new genes involved in the mechanism of action of nitrogen-containing-bisphosphonates used to treat osteoporosis. These drugs impair osteoclast function by inhibiting the mevalonate pathway and protein prenylation. The authors now identify the orphan membrane transporter SLC37A3 as a novel key factor, unrelated to the mevalonate pathway, for the cellular effects of N-BPs.

SLC37A3 forms a complex with ATRAID, a monotopic membrane protein identified in a companion study. This complex is required for the cellular action of N-BPs, it localizes to lysosomes and it exports N-BPs internalized by endocytosis (or phagocytosis) from the lysosomal lumen to the cytosol, where these drugs exert their effect.

The study is rigorously designed and executed. It provides compelling evidence for the roles of SLC37A3 and ATRAID in the cellular entry of N-BPs, a key mechanism which remained elusive thus far, with potential clinical importance.

Essential revisions:

1) Concerning: "…. SLC37A3, but it is predicted based on sequence homology to be a sugar-phosphate antiporter (Chou et al., 2013)."

The related SLC37 proteins are strictly speaking glucose-6-phosphate: phosphate antiporters. However, SLC37A3 has been tested in similar experimental set ups, and it was found not to be a glucose-6-phosphate: phosphate antiporter (Pan et al., 2011). To the best of our knowledge, its physiological transport specificities have not been determined. It would be important to add this information.

2) Concerning: "… we expressed functional, epitope-tagged SLC37A3 and ATRAID from near-endogenous levels of transcripts (Figure 2—figure supplement 2B-C and Figure 3—figure supplement 1A) and characterized their localization with immunofluorescence (IF)."

It is a little confusing that the authors in the Materials and methods section refer to single rescues with either sATRAID-V5, lATRAID-V5 or SLC37A3-HA as sub-endogenous expression levels and the KO2+SLC37A3-HA with co-expression of sATRAID-V5 or lATRAID-V5 as near-endogenous expression levels, and then in the sixth paragraph of the Results and Discussion generalize and refer to all situations as near-endogenous levels of transcripts. Especially, since more of the results in Figure 3—figure supplement 1A, respectfully, do not look like near-endogenous levels. Indeed, curiously, the expression of SLC37A3-HA in KO2+SLC37A3-HA seems to be downregulated when co-expressed with as well sATRAID-V5 as lATRAID-V5.

The reduced expression of SLC37A3-HA in the KO2+SLC37A3-HA cells co-expressing either sATRAID-V5 or lATRAID-V5 might explain why these cells show a response to high concentrations of alendronate between those of WT cells and KO2 and KO2+SLC37A3-HA cells in Figure 3—figure supplement 1B.

To sum up, it would be preferable just to relate to the actual expression levels observed in Figure 3—figure supplement 1A.

3) Concerning: "In the absence of ATRAID, the mature, glycosylated population of SLC37A3 (around 50kD) became undetectable, whereas a population of un-glycosylated SLC37A3 (around 40kD) emerged (Figure 3G and Figure 3—figure supplement 2F)." And "Figure 3—figure supplement 2F legend, the band corresponding to an unglycosylated population of SLC37A3 that is present in the absence of ATRAID but not in the presence of ATRAID is marked with an asterisk."

The statements do not seem to be consistent with the data in Figure 3F. In Figure 3F "Input IB: HA", the glycosylated form seems to be at the same level with (SLC37A3-HA + ATRAID-V5) and without (SLC37A3-HA +GFP-V5) ATRAID expression, while the unglycosylated form is the most prominent form in both lanes. Please explain why, or modify the interpretation of the results.

---

## [Author Response]

Essential revisions:1) Concerning: "…. SLC37A3, but it is predicted based on sequence homology to be a sugar-phosphate antiporter (Chou et al., 2013)."The related SLC37 proteins are strictly speaking glucose-6-phosphate: phosphate antiporters. However, SLC37A3 has been tested in similar experimental set ups, and it was found not to be a glucose-6-phosphate: phosphate antiporter (Pan et al., 2011). To the best of our knowledge, its physiological transport specificities have not been determined. It would be important to add this information.

The nomenclature has been corrected, and additional discussion on the physiological function of SLC37A3 has been added (Results and Discussion, second paragraph).

2) Concerning: "… we expressed functional, epitope-tagged SLC37A3 and ATRAID from near-endogenous levels of transcripts (Figure 2—figure supplement 2B-C and Figure 3—figure supplement 1A) and characterized their localization with immunofluorescence (IF)."It is a little confusing that the authors in the Materials and methods section refer to single rescues with either sATRAID-V5, lATRAID-V5 or SLC37A3-HA as sub-endogenous expression levels and the KO2+SLC37A3-HA with co-expression of sATRAID-V5 or lATRAID-V5 as near-endogenous expression levels, and then in the sixth paragraph of the Results and Discussion generalize and refer to all situations as near-endogenous levels of transcripts. Especially, since more of the results in Figure 3—figure supplement 1A, respectfully, do not look like near-endogenous levels. Indeed, curiously, the expression of SLC37A3-HA in KO2+SLC37A3-HA seems to be downregulated when co-expressed with as well sATRAID-V5 as lATRAID-V5.The reduced expression of SLC37A3-HA in the KO2+SLC37A3-HA cells co-expressing either sATRAID-V5 or lATRAID-V5 might explain why these cells show a response to high concentrations of alendronate between those of WT cells and KO2 and KO2+SLC37A3-HA cells in Figure 3—figure supplement 1B.To sum up, it would be preferable just to relate to the actual expression levels observed in Figure 3—figure supplement 1A.

We have removed descriptions such as “near-endogenous” or “sub-endogenous”. Instead we emphasize that in rescue experiments we have taken care to avoid over-expression of SLC37A3 and ATRAID (Results and Discussion, sixth paragraph). Readers are referred to Figure 3—figure supplement 1A for specific information on expression levels of SLC37A3 and ATRAID in rescue strains. We chose not to comment on the impact of ATRAID expression on SLC37A3 transcript levels and other minor consequences of lower-than-endogenous expression levels of SLC37A3 (such as those observed in Figure 3—figure supplement 1B) because they are not essential to our main arguments.

3) Concerning: "In the absence of ATRAID, the mature, glycosylated population of SLC37A3 (around 50kD) became undetectable, whereas a population of un-glycosylated SLC37A3 (around 40kD) emerged (Figure 3G and Figure 3—figure supplement 2F)." And "Figure 3—figure supplement 2F legend, the band corresponding to an unglycosylated population of SLC37A3 that is present in the absence of ATRAID but not in the presence of ATRAID is marked with an asterisk."The statements do not seem to be consistent with the data in Figure 3F. In Figure 3F "Input IB: HA", the glycosylated form seems to be at the same level with (SLC37A3-HA + ATRAID-V5) and without (SLC37A3-HA +GFP-V5) ATRAID expression, while the unglycosylated form is the most prominent form in both lanes. Please explain why, or modify the interpretation of the results.

A note has been added in the main text to clarify this apparent discrepancy (Results and Discussion, seventh paragraph). In Figure 3F, the cell lines used for the reciprocal-IP experiment express SLC37A3 at a much higher level compared to the endogenous level. Such over-expression may have saturated the machinery required for SLC37A3 degradation and attenuated the difference in SLC37A3 expression between cell lines expressing SLC37A3 with or without ATRAID. In Figure 3G, all cell lines express SLC37A3 at either the endogenous level or lower-than-endogenous levels (Figure 3—figure supplement 1A). Therefore, the pattern shown in Figure 3G, which is not complicated by artifact caused by over-expression, is more likely to represent the physiological truth.

Indeed, in cells over-expressing SLC37A3 with or without ATRAID, SLC37A3 exists predominantly in the un-glycosylated form. However, the ATRAID dependent glycosylation of SLC37A3 in these cells is still partially preserved. Although glycosylated SLC37A3 molecules can be observed both in cells over-expressing SLC37A3 with and without ATRAID, the specific 50 kD band that corresponds to the major band observed in lane 2, 4 and 5 of Figure 3G is only present in cells co-expressing SLC37A3 and ATRAID. (We do not know the exact identities of those SLC37A3 bands that migrate slower or faster than the 50 kD band.) This pattern is not particularly clear in Figure 3F because samples in Figure 3F were analyzed on 4-20% SDS-PAGE gels, which offer good overall size separation but not specifically at around 50 kD. Other immunoblots against SLC37A3, such as Figure 3G and Figure 3—figure supplement 2F-G, were performed using 10% SDS-PAGE gels, which offer excellent size separation at around 50 kD. We have analyzed the samples used in Figure 3F with the protocol used in Figure 3—figure supplement 2F and confirmed the statements above. We chose not to include this piece of data due to its lack of new information.

In summary, as cell lines used in Figure 3G and Figure 3—figure supplement 2F do not over-express SLC37A3 and more faithfully represent the physiological state, we formulated our interpretation based largely on these two figures.